# Optical Sensing Using Hybrid Multilayer Grating Metasurfaces with Customized Spectral Response

**DOI:** 10.3390/s24031043

**Published:** 2024-02-05

**Authors:** Mahmoud H. Elshorbagy, Alexander Cuadrado, Javier Alda

**Affiliations:** 1Physics Department, Faculty of Science, Minia University, El-Minya 61519, Egypt; 2Grupo Complutense de Optica Aplicada, Departamento de Optica, Facultad de Óptica y Optometría, Universidad Complutense de Madrid, Av. Arcos de Jalón, 118, 28037 Madrid, Spain; javier.alda@ucm.es; 3Escuela de Ciencias Experimentales y Tecnología, Universidad Rey Juan Carlos, 28933 Móstoles, Spain; alexander.cuadrado@urjc.es

**Keywords:** multilayer grating, optoelectronic sensor, surface plasmon resonance, spectral control

## Abstract

Customized metasurfaces allow for controlling optical responses in photonic and optoelectronic devices over a broad band. For sensing applications, the spectral response of an optical device can be narrowed to a few nanometers, which enhances its capabilities to detect environmental changes that shift the spectral transmission or reflection. These nanophotonic elements are key for the new generation of plasmonic optical sensors with custom responses and custom modes of operation. In our design, the metallic top electrode of a hydrogenated amorphous silicon thin-film solar cell is combined with a metasurface fabricated as a hybrid dielectric multilayer grating. This arrangement generates a plasmonic resonance on top of the active layer of the cell, which enhances the optoelectronic response of the system over a very narrow spectral band. Then, the solar cell becomes a sensor with a response that is highly dependent on the optical properties of the medium on top of it. The maximum sensitivity and figure of merit (FOM) are S_B_ = 36,707 (mA/W)/RIU and ≈167 RIU^−1^, respectively, for the 560 nm wavelength using TE polarization. The optical response and the high sensing performance of this device make it suitable for detecting very tiny changes in gas media. This is of great importance for monitoring air quality and thecomposition of gases in closed atmospheres.

## 1. Introduction

Optical sensors that are based on surface plasmon resonance (SPR) provide a label-free, real-time detection technique for many specimens in chemistry [1], biology [2], and environmental sciences [3]. Different designs have been applied to enhance the coupling of the incoming light to the SPR [4,5]. The mechanisms of such devices are purely optical-, phase-, spectral-, angular-, or intensity-interrogated [6,7,8,9,10,11]. A modern simplification of these sensors is obtained by attaching them to photodetectors, which extract the measurable quantity from an electrical signal [12,13]. An optoelectronic sensor is the next generation of this kind that combines both functions in one device by activating the surface of the photodetector to measure variations in the parameters of interest in the surrounding medium [14,15,16]. A further development of these devices is the customization of their performance and range for specific applications. For example, atmospheric quality monitoring (mainly for gas content) involves very tiny changes in the refractive index close to unity with a resolution of $10^{-4}$ RIU [17,18]. Monitoring changes in environments with gases at high pressure or containing organic vapors requires a wide dynamic range to cover their high-value refractive indices compared to the refractive index of air [15,18]. These different situations must be taken into account in the sensor design, while the characteristics of the plasmonic resonance is spatially (long- and short-range SPRs) and spectrally (narrow and wide) controlled [19,20]. Long-range spatial decay with an ultra-narrow spectral feature can sense variations with a high resolution [21,22,23].

An approach to converting a solar cell into a plasmonic sensor begins by deactivating the direct transmission of light into the cell by replacing the transparent top contact by a metallic thin film. This step makes the optoelectronic response of the device almost zero. Then, we generate an optoelectronic response by a plasmonic resonance that has the ability to penetrate the cell active layer. These two changes in the design functionalize the cell surface to be sensitive to small changes in the index of refraction of the surrounding medium. A grating with a subwavelength period removes the higher-order diffraction modes. This makes the response as simple as possible. A metallic grating can achieve the job. However, within the visible band, where most of the solar cells are operative, metals are very good reflectors and absorb the rest of the light within their skin depth. This reduces the amount of light that can reach the cell active layer. Therefore, a better option involves lossless materials such as dielectrics.

The functionalization of a design to work with an electronic interrogation mechanism requires the investigation of both the sensing response and the signal conversion method. However, the integration of a sensing mechanism with a state-of-the-art electronic device is a short way to achieve this goal. In this case, adding a metasurface on top of a fabricated solar cell can functionalize it as an efficient sensor. The design starts with a solar cell that can be suitable for sensing applications. For the proper integration of the device with other systems, it has to be lightweight with a small volume. These conditions lead to thin-film solar cells [24,25]. Then, for compatibility with fabrication methods involving thermal treatments, we go to the stable inorganic thin-film solar cell type [26,27]. Lastly, when considering low-cost and abundant nontoxic materials, the hydrogenated amorphous silicon (aSiH) thin-film solar cell is a good selection for our design [28,29,30]. The performance of solar cells based on aSiH is improved when thermally treated [31,32]. In the case of ambient monitoring, the response of the cell has to be sensitive to very small variations in the surrounding medium. Previous attempts have shown how this design works and adapted one-layer gratings or high-aspect-ratio gratings to convert the cell surface into a sensor [17,18,33]. The sensitivity and measuring range significantly change with the design parameters, with the refractive index contrast being the main parameter in the design, besides the shape and geometry.

The goal of this study is to select suitable materials and optimize the geometrical parameters. With this, we couple the cell response to changes in the cell surface. Hybrid multilayer gratings made of dielectric materials with different refractive indices can provide an optimal matching and smooth refractive index contrast by selecting their refractive indices according to the refractive index of the surrounding medium. This adds one more degree of freedom to the design parameters compared with the single-layer grating option. Here, as we aim to monitor changes in gases with refractive indices close to that of air (n≃1), a combination of transparent low- and high-refractive-index materials is a good choice. Then, we have to determine the best arrangement and geometrical parameters of the materials to obtain the best performance of the device as a sensor. The nanostructured design based on the multilayer subwavelength structure generates a plasmonic resonance that enhances the absorption of light by the active layer, which generates an increase in the deliverable electric current. Our choice of materials for the multilayer grating, along with the optimization of their geometrical parameters, produces a refractometric sensor with high sensitivity.

After this Introduction Section, we present how the architecture of the aSiH solar cell can be re-arranged to work as an optical sensor in Section 2. In this section, we also describe the main ideas behind the optimization process that provides our final design. This is carried out through computational electromagnetism, where a dedicated merit function is defined in terms of the energy absorbed by the active layer of the structure. Section 3 is then devoted to the presentation of the results obtained from our model and how they can be applied to the case of a refractometric sensor for gases. The performance of the device as a gas sensor and the related sensing parameters are presented in Section 4. Finally, Section 5 summarizes the main results of this contribution.

## 2. Design and Model

The design is based on a thin-film solar cell made of aSiH, and the purpose of the proposed layer structure is to convert it into a refractive index sensor with a narrow response related to the SPR.

### 2.1. Solar Cell with Multilayer Grating

The design is depicted in Figure 1. It starts with the aSiH solar cell architecture [34] presented in Figure 1a. It consists of a bottom metallic contact/zinc oxide buffer layer (ZnO)/n-type aSiH/i-type aSiH/p-type aSiH/top contact. The geometrical parameters and refractive index data sources are listed in Table 1, and the value used for glass (SiO_2_) is 1.47. The legend in Figure 1a defines the materials for each layer. The sensing mechanism is enabled by the plasmonic effect occurring on top of the solar cell. The performance of plasmonic sensors is characterized by the sensitivity, figure of merit (FOM), dynamic range, etc. For a refractometric sensor, these parameters depend on the variation in the measured variable when changing the refractive index.

Our proposal is based on an existing and proven technology: the amorphous silicon solar cell. This approach eases the development of the sensor, as it involves few changes to the fabrication process. For this, we will optimize only the structure responsible for generating the plasmonic effect to produce the highest signal possible from a conventional solar cell. As we are not interested in using the device for energy-harvesting applications, the transparent top contact in conventional aSiH solar cells is replaced by a thin metal film made of silver with a thickness of 50 nm. This layer prevents the direct transmission of light to the cell active layer, so the response is almost null. The selection of silver is based on its strong, sharp plasmonic response when compared with other metals [41]. However, silver surfaces can oxidize easily. To avoid this degradation, the whole structure is coated with a very thin layer of SiO_2_ ($t_{SiO_{2}}$ = 10 nm), which passivizes the silver layer without changing the optical response. Now, to reactivate the response of the cell for a specific wavelength, we need to add the multilayer grating on top of the silver thin-film layer. Once we do that, and with the proper selection of materials and geometry, we can activate the SPR that propagates to the active layer and creates a response related to it (see Figure 1b). This step functionalizes the solar cell response, which becomes a sensor for detecting changes in the refractive index of the surrounding medium. The multilayer grating is made of three layers of, arranged from bottom to top, magnesium fluoride (MgF_2_, refractive index ≈ 1.38 @ λ = 560 nm)/silicon nitride (SiN_3_, refractive index ≈ 2 @ λ = 560 nm)/magnesium fluoride. These materials, MgF_2_ and SiN_3_, are widely used in the optical designs of antireflection coatings and diffraction gratings [42]. These materials also provide a refractive index contrast with a low-index surrounding medium, which is air ($n_{g}≃1$) or a gas at atmospheric pressure, which has a refractive index value close to it.

The geometrical parameters of the grating are the width (GW), height (GH), and period (P), which will be optimized for the selected operating wavelength. The height of SiN_3_ will be fixed at 2 × GH, while that of MgF_2_ will be GH. To make the system independent of the optical power of the light source, the response of the system will be in terms of responsivity $R_{S}$ = output current/input power (see Figure 1c).

### 2.2. Optical Model and Merit Function

The optical model was built with the simulation package COMSOL Multiphysics version 6.1 wave optics module, which is based on the finite element method. Our structure is an extruded 1D grating that can be modeled using a 2D simulation. The computed geometry is finite and periodic in one direction (*X*-axis) and infinite along the extruded direction (*Z*-axis). On top of the layer structure in Figure 1a, we place an optical source defined by an electric field of linear polarization with an amplitude E = (0,0,1) V/m for the TE mode (the TM mode is excited using a magnetic field vector with an amplitude H = (0,0,1) A/m). The source is modeled as a periodic port that is suitable for symmetric arrays. In this port, we internally calculate the diffraction orders, if any, as a function of the source wavelength and period of the grating. As we have only normal incidence and the grating is subwavelength, the port with the automated diffraction-order calculation absorbs the backward waves from the structure, and there is no need for PML or scattering boundary conditions. The meshing uses triangular elements. The COMSOL package can solve Maxwell’s equations for the fields when every element is at least five times smaller than the wavelength. For better accuracy, we used a maximum element size of λ/10, increasing the meshing density when required. On both sides of the unit cell in Figure 1a, we define periodic boundary conditions to account for the infinite array of the structure.

The irradiance of the light source that illuminates the device is 50 mW/cm^2^. This is the typical irradiance of conventional diode lasers. The typical line width of a commercial laser diode is well below 0.1 nm. So, if the spectral width of the resonance is a few nanometers, the light source can be considered monochromatic, which is the case for electronic and intensity interrogation.

For simplicity, in the optimization process, we will use the absorption in the active layer of the cell. The absorption is a function of the square of the electric field amplitude and the complex optical constants of the active layer according to the following equation [43]:(1)$$P\left(\omega \right)=\frac{1}{2}\omega \varepsilon^{\prime \prime }\left(\omega \right){\left|E\left(\omega \right)\right|}^{2},$$
where |E(w)| is the modulus of the electric field, $\epsilon^{\prime \prime }$ is the imaginary part of the dielectric permittivity of the medium through which the electric field propagates, and ω is the angular frequency of the electromagnetic radiation. P(ω) in Equation (Equation 1) is evaluated locally and integrated within the volume of the element of interest. This absorbed power is normalized to the input power to give the absorptance, which we will use as a merit function to maximize. The optimization parameters will be the wavelength of the incident light and the GH and GW of the grating.

Once the incident light is absorbed by the active layer of the cell, it generates a photocurrent $I_{\mathrm{sc}}$. This current is the output signal of the sensing device. In this case, the change in the index of refraction of the surrounding medium will change that output current, leading to a purely electronic interrogation mechanism of the device. To make the device independent of the optical power of the illumination source, the responsivity, $R_{S}$, is the parameter of interest for the measurements (see Figure 1c). $R_{S}$ is related to the output current and the incident optical power through the following equation [44]:(2)$$R_{S}\left(n_{g}\right)=\frac{I_{\mathrm{sc}}\left(n_{g}\right)}{P_{\mathrm{input}}},$$
where $I_{\mathrm{sc}}$ is the maximum photo-generated current, and $P_{\mathrm{input}}$ is the incident optical power of the illumination source.

## 3. Results and Discussion

The design of an optimal multilayer grating involves the choice of different materials and geometrical parameters, which affect the optical response of the whole design. As we already fixed the materials to those conventionally used in antireflection coatings, the rest of the process involves optimizing the geometrical parameters of the grating, such as the period (P), GH, and GW, and the operating wavelength. An aSiH solar cell has an optoelectronic response in the visible range (for wavelengths between 350 and 700 nm) in terms of the active absorption. By active absorption, we mean the absorption that occurs in the active layer of the cell (iaSiH, in our case) and can be converted into an electrical current. Then, the operating wavelength must lie within this range. Usually, increasing the operating wavelength of a plasmonic sensor increases the sensitivity at the cost of the dynamic range of the device [45]. In this case, we have selected two wavelengths of interest, namely, 430 nm and 560 nm, for short and long wavelengths, respectively, and will show the different options for both the sensitivity and dynamic range. This selection can be any inside the mentioned range where the cell is responsive.

The periods are fixed at 400 nm for a wavelength of 430 nm and 530 nm for a wavelength of 560 nm so as to have only zero-order diffraction, as the structure is subwavelength in size. The optimization is applied to both GH and GW to tune the resonance and increase the optical power toward the active layer for a narrow band centered around the selected wavelengths. The narrow-band characteristic is favorable for sensing applications due to being highly dependent on the surrounding medium.

The results of this optimization are shown in Figure 2. As we see, several combinations of GW and GH reveal a high absorption in the active layer. The maximum of each line is highlighted with white and red circles. The global maximum of this active absorption for each line is highlighted by the red circles and marked as 1,2,3,4, and the results are summarized in Table 2. Although the total absorption is close to unity (perfect absorption), it is shared with the metallic layer and p-aSiH for the 430 nm wavelength and shared with the metallic layer only for the 560 nm wavelength. The plasmonic resonance generated on the surface of the metallic top contact confines and enhances the optical field, allowing it to penetrate the active layer and thus making this absorption sensitive to environmental changes, as it is linked to the plasmonic resonance.

The optical absorption for the optimized cases is shown in Figure 3a,b for 430 nm and 560 nm wavelengths, respectively, and using two different polarization states. In these plots, we see how the plasmonic resonances efficiently confine light and couple it to the bottom cell layers, resulting in a perfect absorption effect. The line widths at resonances for the 430 nm case are δλ=3.31 nm in the TE case and δλ=3.39 nm in the TM case. The line widths are much narrower for the 560 nm wavelength resonance: they are δλ=1.91 nm in the TE case and δλ=2.8 nm in the TM case. These narrow lineshapes are of great importance when applying the design for sensing, as the response will be highly sensitive to small variations in the surrounding medium characteristics. Based on how the delivered signal for each case changes with the refractive index of the surrounding medium, we can know the potential applications for each case.

Plasmonic resonance is responsible for achieving strong light confinement close the resonant structure. In Figure 4, we represent the electric and magnetic field distributions for the TE and TM cases, respectively. Each map is defined for the optimal cases provided in Figure 2. They correspond to the selected wavelengths and polarization states expressed in the figure. As we can see, the high-field surface plasmon resonance activated on top of the metallic layer penetrates well into the cell active layer. This helps to generate a photocurrent signal caused by the excitation of plasmonic resonances.

The power loss function described in Equation (Equation 1) is normalized (Normalized $Q_{\mathrm{Loss}}$) to the input power and visualized in Figure 5. The longer the wavelength, the deeper the penetration into the active layer [46,47,48]. However, the absorbed power is similar in all cases, with the total absorbed power for TE cases being higher than that for TM cases for both wavelengths. In Figure 4 and Figure 5, the two maps on the right have a different field distribution in the active layer that is wave-like. This happens because the thickness of the device is small (≈500 nm), and the surface plasmon waves reach the bottom metallic contact, which acts as a mirror, with no negligible amplitude [49]. Therefore, interference effects are responsible for this kind of field distribution. This distribution is distinguished from the incident plane wave at the top domain before reaching the structure. The total absorbed power is obtained by integrating the normalized power loss function over the whole area of the active layer and is reported in Table 2. In the next section, we will check how the high field confinement located on top of the device and the high optical absorption happening in the active layer of the cell contribute to the performance of the device as a sensor.

## 4. Gas Sensing with High Resolution

The increase in the refractive index of a medium modifies the way that light propagates through it. Furthermore, if this medium is in contact with a region with high optical field confinement (plasmonic resonance, for example) that decays fast in the spatial domain, this interface property will be highly sensitive to any variations in this medium. These two important phenomena already exist in our design, which makes it suitable for sensing changes in the refractive index of a medium. As we performed our optimization with the top medium as a vacuum, this design will be suitable for gas media that have a refractive index close to unity. However, it can be redesigned through the geometrical parameters to other media, for example, liquids.

To check the performance of the device as a gas refractive index sensor, we studied the changes in the optoelectronic response of the device. This is simply the change in the responsivity of the device with respect to the surrounding medium’s refractive index change, $n_{g}$. The sensitivity of this device is defined as [50,51]
(3)$$S_{B}=\frac{\partial R_{S}}{\partial n_{g}},$$

For intensity- and electronic-modulated sensors, the figure of merit (FOM) is defined in terms of the sensitivity and the responsivity or intensity [50,51]:(4)$$\mathrm{FOM}=\frac{S_{B}}{R_{S}}.$$

This parameter represents the resolution of the sensor and its ability to distinguish between two measurements with a tiny variation in the measured parameter. The results are shown in Figure 6, where the responsivity decays fast when increasing the refractive index of the top medium. The power of the illumination source is fixed for the whole study at 50 mW/cm^2^, which is typical for laser diodes. The photocurrent is obtained after integrating the optical absorption over the volume of the active layer, including the surface area of the cell. This approach normalizes the value of the current to the input. Moreover, the responsivity is defined as the ratio between the delivered signal and the input optical power, canceling the dependence on the power of the optical source. The best sensitivity was obtained for the TE polarization case, with similar values for TM polarization. The average sensitivity using TE polarization is S_*B*_ = 13,320 (mA/W)/RIU for the 430 nm wavelength in the range of n_*g*_ 1.001–1.005 RIU. The maximum responsivity in this case is 168.36 mA/W, which gives an FOM of ≈ 79 RIU^−1^. This is suitable for measuring refractive index changes in organic vapors or gases at high pressure in closed atmospheres, where the refractive index change is relatively large compared with the value of the vacuum. The other case treated in this paper shows an average sensitivity of S_*B*_ = 36,707 (mA/W)/RIU for the 560 nm wavelength in the range of n_*g*_ 1.001–1.005 RIU. The maximum responsivity in this case is 219.85 mA/W, revealing an FOM of ≈167 RIU^−1^. The sharp decay of the responsivity and the high values of sensitivity and FOM make the device at this wavelength suitable for sensing pressure leakage in ultrahigh-vacuum systems and for the monitoring of the air quality in closed atmospheres.

In both cases, we have found a high sensitivity of the device to detect changes in the content or pressure of a closed atmosphere, which is competitive with recent approaches [52,53,54,55,56,57,58,59]. The system operates under normal incident conditions and can be micro-sized, which implies a compact platform for gas-sensing applications that can be integrated easily into different systems. The deliverable signal of the device further eases its operation by excluding the complex spectrometers and goniometers used in spectral and angular interrogation mechanisms. The dependence of the sensing parameters is not affected significantly by different polarizations but is greatly affected by the operating wavelength. The longer the wavelength, the higher the sensitivity; as commonly happens for plasmonic devices [45], the range is limited here to the operating spectrum of the solar cell itself (in the case of aSiH solar cell, it is 350–700 nm).

## 5. Conclusions

In this contribution, we modified the typical layer arrangement of a hydrogenated amorphous silicon solar cell to have it work as a refractometric sensor for gases. The main idea relies on the addition of a metasurface made of hybrid multilayer dielectric gratings. This multilayer subwavelength structure generates plasmon resonances that propagate through a metallic silver layer placed on top of the classical solar cell structure, reaching the active layer and generating an electric current. The materials, thicknesses, and width of the multilayered grating were chosen and dimensioned to increase absorption by the active layer of the device.

After selecting two wavelengths of operation, we obtained the geometries providing the maximum value for the absorption of light. This analysis was performed for TE and TM polarization, finding quite similar behavior in both cases. An interesting result of these structures is that the optical absorption for the TM mode is a little wider than for the TE case. When evaluating how the system works as a refractometric sensor, we observed that the responsivity is slightly larger for the TE mode while maintaining a similar value of sensitivity for both modes. Our results show that a refractometric sensor with the parameters GH = 224 nm and GW = 70 nm working in the TE mode at λ=560 nm works the best, with a sensitivity value as large as $3.67\times 10^{4}$ (mA/W)/RIU and an FOM of ≈ 167 RIU^−1^. In summary, we have demonstrated how a well-established technology can be easily modified to serve as an optical sensor. This modification is based on the use of specifically tailored metasurfaces.

## Figures and Tables

**Figure 1 sensors-24-01043-f001:**
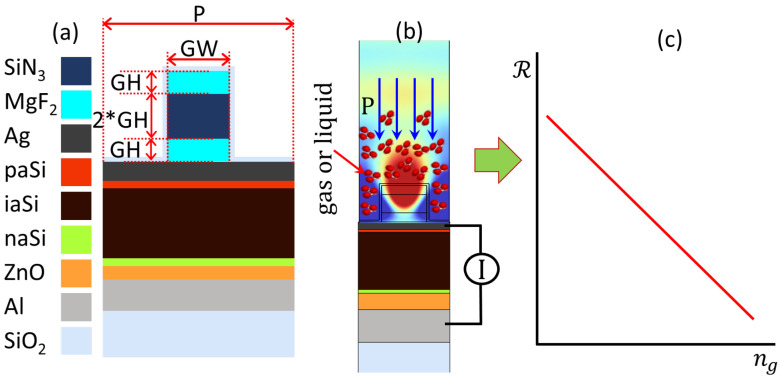
(**a**) The layer structure of the aSiH solar cell with the top contact made of silver and a metasurface on top; (**b**) the operation mechanism of the device, (**c**) the electronic output of the sensor.

**Figure 2 sensors-24-01043-f002:**
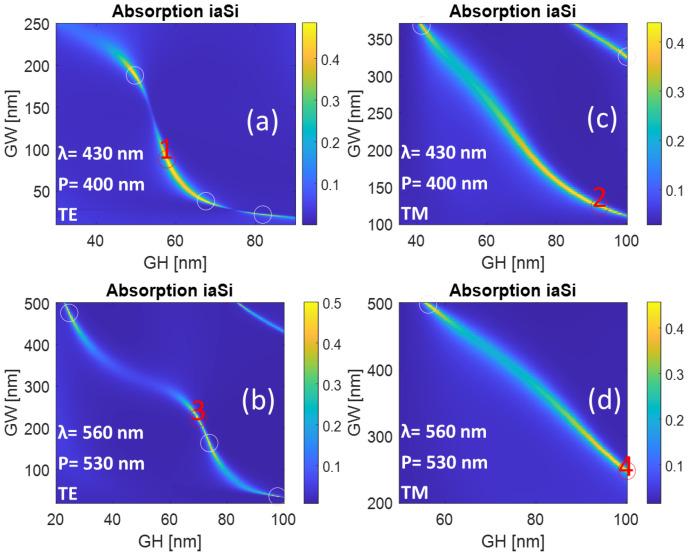
The absorption in the active layer as a function of the multilayer grating width and height for (**a**,**c**) a wavelength of 430 nm and (**b**,**d**) a wavelength of 560 nm. The numbers in red correspond to the designs presented in Table 2.

**Figure 3 sensors-24-01043-f003:**
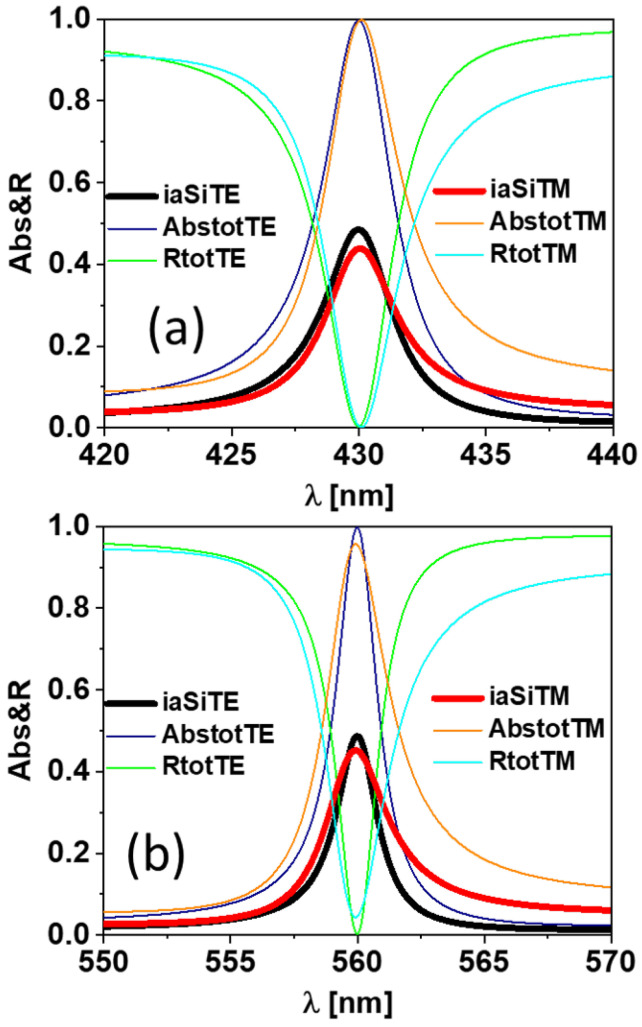
(**a**) The spectral response of the device for a wavelength of 430 nm, TE and TM; (**b**) the spectral response of the device for a wavelength of 560 nm, TE and TM.

**Figure 4 sensors-24-01043-f004:**
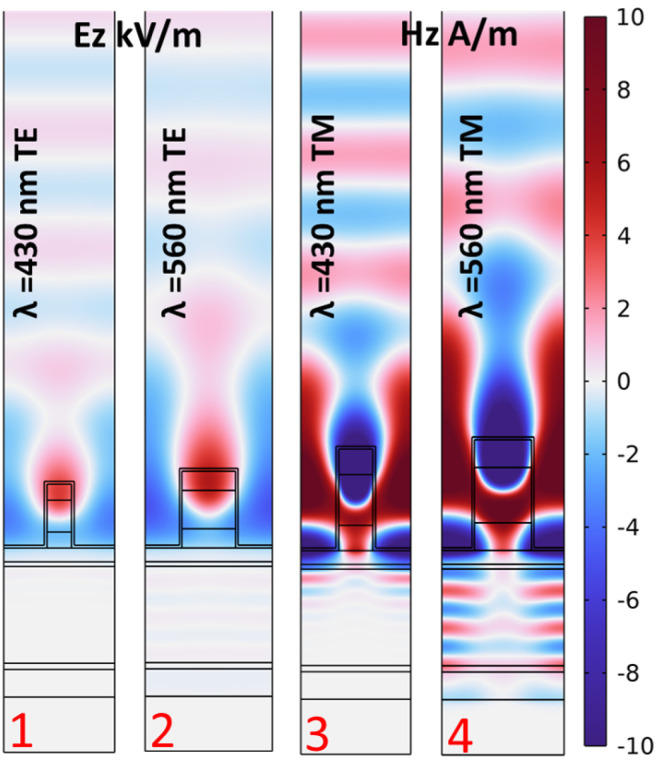
Electric and magnetic field distributions at resonance wavelengths for the cases highlighted by red numbers in Figure 2. The wavelength and polarization type of each case are presented at the top of each map, while the case number is at the bottom of the map.

**Figure 5 sensors-24-01043-f005:**
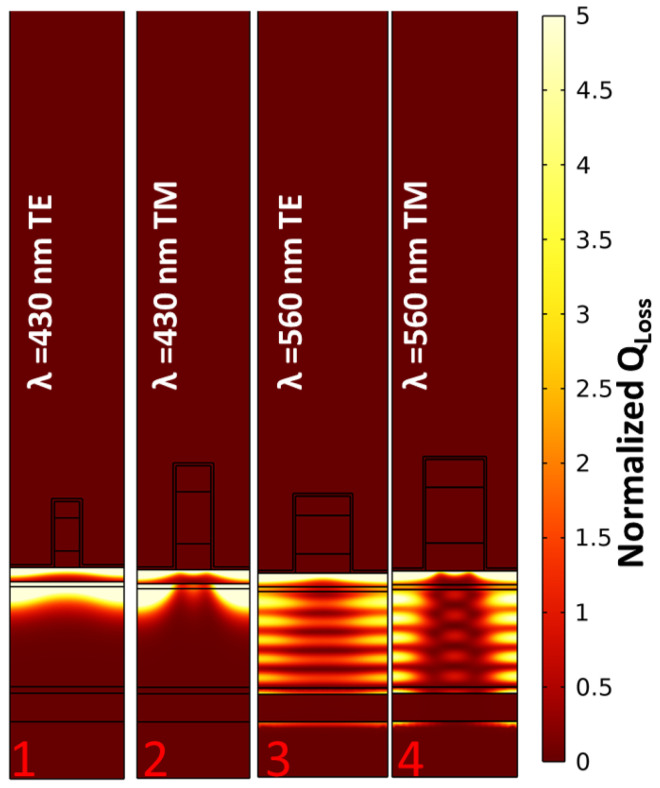
The spatial distribution of the normalized dissipated energy at the resonance wavelengths for the cases highlighted by red numbers in Figure 2. The wavelength and polarization type of each case are presented at the top of each map, while the case number is at the bottom of the map.

**Figure 6 sensors-24-01043-f006:**
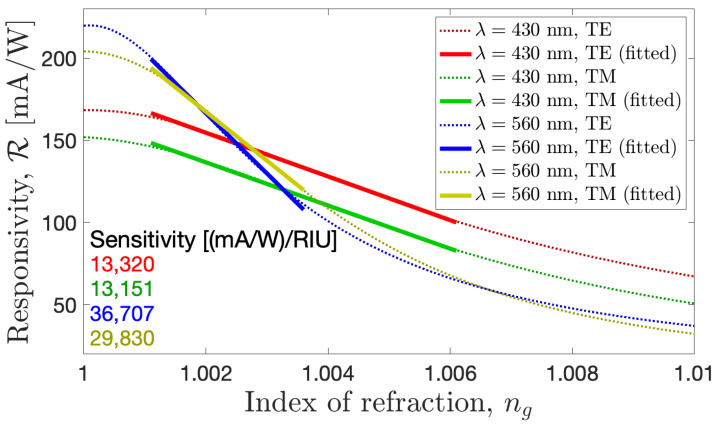
The responsivity of the device as a function of the refractive index change in the top medium for the selected optimized cases.

**Table 1 sensors-24-01043-t001:** Geometrical parameters and layer arrangement of the device.

Layer	Shape and Geometry [nm]	Materials and Ref.
Substrate	Infinite layer	Glass
Back contact	Thin film [200]	Aluminum [35]
Buffer layer	Thin film [100]	Aluminum-doped zinc oxide [36]
Buffer layer	Thin film [22]	n-type hydrogenated amorphous silicon [37]
Active layer	Thin film [350]	i-type hydrogenated amorphous silicon [37]
Buffer layer	Thin film [17]	p-type hydrogenated amorphous silicon [37]
Top contact	Thin film [50]	Silver [38]
Metasurface	Three-layer grating, GW, GH, P	MgF_2_/Si_3_N_4_/MgF_2_ [39,40]
Protecting layer	Thin film [10]	SiO_2_

**Table 2 sensors-24-01043-t002:** Optimized widths and heights for the selected cases.

Maximum Number- Wavelength (nm)	Polarization	GW, GH [nm]	Maximum Absorption
1–430	TE	88, 58	0.485
2–430	TM	124, 92	0.438
3–560	TE	224, 70	0.486
4–560	TM	248, 100	0.45

## Data Availability

All the data of this study are available and included within the manuscript itself.

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
