# Peer review of "Optical Sensing Using Hybrid Multilayer Grating Metasurfaces with Customized Spectral Response"

_sensors, 2024, doi:10.3390/s24031043_

Round 1
Reviewer 1 Report
Comments and Suggestions for Authors
In this study, the authors propose a design involving a thin film solar cell combined with a metasurface, crafted as a hybrid dielectric multilayer grating for refractive index sensing. The Finite Element Method (FEM) is employed through COMSOL Multiphysics simulations, showcasing a high level of expertise. While the results of this work are intriguing, there is a need for improved explanations of the mechanism and more comprehensive references. Therefore, I recommend accepting this manuscript following a thorough revision.
1. Why was silver selected as the plasmonic material? As commonly acknowledged, silver is favored for its susceptibility to oxidation. Please briefly elucidate in the text (no need to present the simulation results) how the optical properties would differ if silver were substituted with another stable plasmonic material (e.g., gold).
2. For the convenience of readers, please provide details on COMSOL settings, including parameters such as mesh size, port settings, and clarify whether a 2D or 3D model was used (and explain the rationale if 2D is utilized instead of 3D). Additionally, elucidate on the choice of either Scattering Boundary Condition or Perfectly Matched Layer (PML) in the text.
3. The color scheme employed to represent materials in Fig. 1(a) poses challenges in differentiation. I suggest enhancing the color contrast for improved clarity.
4. Aside from the height of SiN3 and MgF2, Figure 1 does not illustrate the heights of other multi-layer materials. Kindly annotate them in both the figure and the accompanying text.
5. To enrich the introduction, include references to additional works related to plasmonic refractive index-based sensors, such as "Scientific Reports, 2021, 11(1), 18515," and "Chinese Journal of Physics, 2021, 71, 286–299."
6. Further elaboration on the mechanism mentioned in lines 194-195 is needed, particularly regarding the statement "The longer the wavelength, the deeper the penetration into the active layer." Alternatively, cite relevant references to enhance clarity on this aspect.
7. If feasible, conduct a comparative analysis (or table) of the obtained results with those from other analogous metasurfaces reported in recent literature.
8. The resolution of the designed sensor should be indicated in the text.
Reviewer 2 Report
Comments and Suggestions for Authors
The topic is related to the metasurfaces in optical devices for sensing applications, particularly in the context of a hybrid dielectric multilayer grating combined with a thin film solar cell. The design aims to enhance the response of the system for improved sensing performance, especially in detecting changes in gas media.
Several noteworthy results may be highlighted:
1. The addition of a multilayer subwavelength structure, generating plasmon resonances, enhances the absorption of light at the active layer, resulting in increased electric current generation.
2. Through careful selection of materials and dimensions, a refractometric sensor demonstrates high sensitivity.
Despite these achievements, there are aspects where the authors' explanations require improvement.
Several key points should be addressed for greater clarity and understanding:
1. In Figure 4, wave-like colored regions are visible: in subfig. 3 at the top and 4 at the bottom. It is necessary to explain the emergence of these "waves" in TM polarization. Similarly, an explanation is required for the "waves" in Figure 5 at the bottom at 560 nm.
2. The study explores the dependence of the gas sensor sensitivity on the dimensions of the upper layer, incident radiation power, and wavelength. However, some researchers may be interested in the optimization of all layers, rather than commencing with a standard structure of a solar cell. In the conclusion, it is imperative to include, a qualitative discussion addressing the question of optimizing the entire sensor structure.
3. On page 9, it is stated, "the longer the wavelength, the higher the sensitivity." However, on page 6, the highest sensitivity is achieved at a shorter wavelength of 430 nm. It is necessary to provide a plausible explanation for this discrepancy or remove the mentioned statement.
4. What version of COMSOL Multiphysics has been employed?
After implementing these minor revisions, the manuscript will be suitable for publication in MDPI Sensors.
Reviewer 3 Report
Comments and Suggestions for Authors
This study is interested to the readers. However, the following issues should be addressed:
1. Keywords:
“Optoelectronic sensor” is not suitable as Keywords in this manuscript because it is not presented on Title and Abstract of the submitted manuscript.
2. Abstract:
The authors claimed that “The maximum sensitivity and FOM are SB=36707 (mA/W)/RIU and ≈ 167 RIU−1, respectively, for the 560 nm wavelength using TE polarization”. Please give the full name of FOM for readers to understand the meaning of FOM.
3. Please give the novelty of this study on the Introduction section.
4. Pages 3 of 12: The authors claimed that “the legend in figure 1,a defines the materials for each layer”. Please revise as “The legend in figure 1,a defines the materials for each layer”.
5. Pages 6 of 12, Table 2: The authors claimed that “maximum number-Wavelength”. Please revise as “Maximum number-Wavelength (nm)”.
6. Page 8 of 12, line 223-224” The authors claimed that “The maximum responsivity for this case is 168.36 mA/W that gives a FOM ≈ 79 RIU−1”. The authors should be given the information that the illuminated power and the area of tested device for readers to evaluated the responsivity.
7. Conclusion:
The authors claimed that “Our results show that a refractometric sensor with parameters GH=224 nm and GW = 70 nm working in the TE mode at λ = 560 nm works the best, with a sensitivity value as large as 3.67 × 105(mA/W)/RIU and a FOM 167 RIU−1”. Please check carefully the correctness of sensitivity value of 3.67 × 105 (mA/W)/RIU. However, the value of sensitivity presented in Abstract and Results & discussion is only 36707 (mA/W)/RIU which less than one order.
Comments on the Quality of English LanguageThis study is interested to the readers. However, the following issues should be addressed:
1. Keywords:
“Optoelectronic sensor” is not suitable as Keywords in this manuscript because it is not presented on Title and Abstract of the submitted manuscript.
2. Abstract:
The authors claimed that “The maximum sensitivity and FOM are SB=36707 (mA/W)/RIU and ≈ 167 RIU−1, respectively, for the 560 nm wavelength using TE polarization”. Please give the full name of FOM for readers to understand the meaning of FOM.
3. Please give the novelty of this study on the Introduction section.
4. Pages 3 of 12: The authors claimed that “the legend in figure 1,a defines the materials for each layer”. Please revise as “The legend in figure 1,a defines the materials for each layer”.
5. Pages 6 of 12, Table 2: The authors claimed that “maximum number-Wavelength”. Please revise as “Maximum number-Wavelength (nm)”.
6. Page 8 of 12, line 223-224” The authors claimed that “The maximum responsivity for this case is 168.36 mA/W that gives a FOM ≈ 79 RIU−1”. The authors should be given the information that the illuminated power and the area of tested device for readers to evaluated the responsivity.
7. Conclusion:
The authors claimed that “Our results show that a refractometric sensor with parameters GH=224 nm and GW = 70 nm working in the TE mode at λ = 560 nm works the best, with a sensitivity value as large as 3.67 × 105(mA/W)/RIU and a FOM 167 RIU−1”. Please check carefully the correctness of sensitivity value of 3.67 × 105 (mA/W)/RIU. However, the value of sensitivity presented in Abstract and Results & discussion is only 36707 (mA/W)/RIU which less than one order.
Round 2
Reviewer 1 Report
Comments and Suggestions for Authors
The authors have adequately replied to my previous questions and comments. I agree that this manuscript can be accepted for publication.
Author Response
Thanks
Reviewer 3 Report
Comments and Suggestions for Authors
This manuscript reports that optical sensing using hybrid multilayer grating metasurfaces with customized spectral response. This study is interested to the readers.
The authors have addressed my concerns and the manuscript was revised accordingly.
Comments on the Quality of English LanguageThe keyword changes from Optoelectronic sensor to Optoelectronic response?
Author Response
This is one of the changes that we already did in the revision.